# Proline-specific aminopeptidase P prevents replication-associated genome instability

Nicola Silva[1,2], Maikel Castellano-Pozo[1], Kenichiro Matsuzaki[3], Consuelo Barroso[1], Monica Roman-Trufero[1], Hannah Craig[4], Darren R. Brooks[5], R. Elwyn Isaac[4], Simon J. Boulton[3], Enrique Martinez-Perez[1,6]*

1 Medical Research Council London Institute of Medical Sciences, London, United Kingdom, 2 Department of Biology, Faculty of Medicine, Masaryk University, Brno, Czech Republic, 3 The Francis Crick Institute, London, United Kingdom, 4 School of Biology, University of Leeds, Leeds, United Kingdom, 5 School of Science, Engineering and Environment, University of Salford, Salford, United Kingdom, 6 Institute of Clinical Sciences, Faculty of Medicine, Imperial College London, United Kingdom

* enrique.martinez-perez@imperial.ac.uk

**Data Availability Statement:** All relevant data are within the manuscript and its Supporting Information files.

## Abstract

Genotoxic stress during DNA replication constitutes a serious threat to genome integrity and causes human diseases. Defects at different steps of DNA metabolism are known to induce replication stress, but the contribution of other aspects of cellular metabolism is less understood. We show that aminopeptidase P (APP1), a metalloprotease involved in the catabolism of peptides containing proline residues near their N-terminus, prevents replication-associated genome instability. Functional analysis of *C. elegans* mutants lacking APP-1 demonstrates that germ cells display replication defects including reduced proliferation, cell cycle arrest, and accumulation of mitotic DSBs. Despite these defects, *app-1* mutants are competent in repairing DSBs induced by gamma irradiation, as well as SPO-11-dependent DSBs that initiate meiotic recombination. Moreover, in the absence of SPO-11, spontaneous DSBs arising in *app-1* mutants are repaired as inter-homologue crossover events during meiosis, confirming that APP-1 is not required for homologous recombination. Thus, APP-1 prevents replication stress without having an apparent role in DSB repair. Depletion of APP1 (XPNPEP1) also causes DSB accumulation in mitotically-proliferating human cells, suggesting that APP1's role in genome stability is evolutionarily conserved. Our findings uncover an unexpected role for APP1 in genome stability, suggesting functional connections between aminopeptidase-mediated protein catabolism and DNA replication.

## Author summary

The accurate duplication of DNA that occurs before cells divide is an essential aspect of the cell cycle that is also crucial for the correct development of multicellular organisms. Mutations that compromise the normal function of the DNA replication machinery can lead to the accumulation of replication-related DNA damage, a known cause of human disease and a common feature of cancer and precancerous cells. Therefore, identifying factors that prevent replication-related DNA damage is highly relevant for human health.

**Funding:** Research in NS lab is funded by the Grant Agency of Czech Republic (GA20-08819S) and a "Start-Up" grant from the Department of Biology of Masaryk University. Funding to REI and DRB was provided by BBSRC grant 24/S12813. Research in the DNA damage response laboratory of SJB is funded by Cancer Research UK, The Francis Crick Institute and by a European Research Council (ERC) Advanced Investigator Grant (RecMitMei). Research in the EM-P laboratory was funded by a BBSRC David Phillips Fellowship (EM-P) and a MRC core-funded grant (MC-A652-5PY60). The funders had no role in study design, data collection and analysis, decision to publish, or preparation of the manuscript.

**Competing interests:** The authors have declared that no competing interests exist.

In this manuscript, we identify aminopeptidase P, an enzyme involved in the breakdown of proteins containing the amino acid Proline at their N-terminus, as a novel factor that prevents replication-related DNA damage. Analysis of *C. elegans* nematodes lacking aminopeptidase P reveals that this protein is required for normal fertility and development, and that in its absence proliferating germ cells display DNA replication defects, including cell cycle arrest and accumulation of extensive DNA damage. We also show that removal of aminopeptidase P induces DNA damage in proliferating human cells, suggesting that its role in preventing replication defects is evolutionarily conserved. These findings uncover functional connections between aminopeptidase-mediated protein degradation and DNA replication.

## Introduction

The faithful replication and transmission of genomic information during the mitotic and meiotic cell division programs is an essential aspect of life. Since the integrity of DNA is constantly challenged by internal and external factors that induce different types of lesions, cells have evolved lesion-specific DNA repair mechanisms that use enzymes to restore genome integrity. However, under conditions of genotoxic stress the repairing capability of these mechanisms can be overwhelmed, resulting in genome instability [1]. Among the different types of DNA lesions, double-strand breaks (DSBs) represent the most toxic form of DNA damage as they can lead to genome rearrangements or chromosome breakage if left unrepaired [2]. In mitotically-proliferating cells, DNA replication is the main source of spontaneous DSBs and replication stress is a major cause of genome instability [3]. Moreover, mutations in the replication machinery are responsible for a plethora of genetic diseases [4].

In meiotic cells, DSBs are deliberately induced by the SPO11 topoisomerase-like protein to initiate the process of meiotic recombination and accurate DSB repair is essential for the production of viable gametes [5]. There are two main pathways capable of repairing DSBs: homologous recombination, a high-fidelity repair mechanism that requires a template provided by a sister chromatid or homologous chromosome, and non-homologous end-joining (NHEJ), which ligates the ends of a DSB without a template, often leaving insertions or deletions at the breakpoint. Promiscuous use of NHEJ repair under conditions of replication stress or in backgrounds deficient in meiotic crossover formation results in genome instability [6]. Accumulation of DSBs during DNA replication in proliferating germ cells or during meiotic S-phase is particularly deleterious for genome stability, as mutations originated from these lesions have the potential to be transmitted to all cells of the offspring.

Similar to genome stability, maintenance of protein homeostasis is essential for cell and organismal survival. This process involves the regulated turnover of proteins by the ubiquitin proteasome system (UPS) and by peptidases that catalyse cleavage of specific peptide bonds. Aminopeptidases that remove the N-terminal amino acid from polypeptide chains can influence the longevity of proteins by exposing a destabilizing N-terminal residue that directs a protein to the ubiquitin-dependent N-end rule pathway [7]. Such aminopeptidases have critical roles in the establishment of polarity in embryos and during meiosis, although their relevant targets in these developmental processes remain unknown [8,9]. These enzymes can also have important roles in generating free amino acids for biosynthesis and energy production from the peptide products of the proteasome [10]. Peptides and proteins with proline as the second amino acid from the N-terminus are however resistant to hydrolysis by most aminopeptidases due to the conformational constraints conferred by the pyrrolidene ring of proline [11].

Enzymatic cleavage of the N-terminal residue from these proteins requires proline-specific aminopeptidase P (APP), a ubiquitous and conserved enzyme capable of efficient hydrolysis of the X-proline amide bond [12,13]. Animals contain three APP homologues, APP1 is a cytosolic enzyme, whereas APP2 is found on the surface of cells where it has a role in the degradation of extracellular peptides with a penultimate proline, such as circulating mammalian bradykinin [14]. APP3 is structurally related to APP1 and APP2, however, it has a very different substrate specificity. Notably, proline in the penultimate position is not required for APP3 aminopeptidase activity. The enzyme is predominately localised in mitochondria and is responsible for stabilising mitochondrial proteins by removing destabilising amino acids from the N-terminus [15]. Mice and flies lacking APP1 (human XPNPEP1), display embryonic lethality and developmental defects including neurodegeneration [16–18], but the processes that APP1 affects to promote fertility and normal development are not known.

In this study, we have used the nematode *Caenorhabditis elegans* to investigate the physiological roles of APP1, uncovering a key role for APP1 in preventing the accumulation of replication-related DNA damage in germ cells. Despite competence in DSB repair, SPO-11 independent DNA damage accumulates extensively in germ lines of *app-1* mutants, consistent with heightened levels of replication stress. Moreover, human cells depleted of APP1 also show accumulation of spontaneous DNA damage, suggesting that the role of APP1 in preventing genomic instability is evolutionarily conserved.

## Results

### APP-1 promotes normal viability and prevents the accumulation of recombination-independent DNA damage in germ cells

To determine if APP1 is required for normal development in *C. elegans*, we assessed embryonic lethality and morphological defects among the progeny of worms carrying two different *app-1* mutant alleles: one carrying a large in-frame deletion (*app-1(tm1715)*) and a second (*app-1(fq96)*) carrying a near complete deletion of the *app-1* gene created by CRISPR (Fig 1A). Progeny from *app-1(tm1715)* and *app-1(fq96)* mutants displayed reduced brood size, high levels of embryonic lethality, and the presence of developmental defects amongst viable larvae, which are manifested by the presence of worms with abnormal body shapes, sizes, or movement (Fig 1B). Thus, APP-1 is required for viability and normal growth in *C. elegans* as previously observed in mice lacking the APP-1 mammalian orthologue XPNPEP1 [16].

The phenotypes observed in *app-1* mutants are also reminiscent of *C. elegans* mutants that display deficient DNA repair [6,19], leading us to investigate if *app-1* mutant germ lines accumulated DNA damage. Accurate DNA repair in the germ line is essential not only to prevent the accumulation of mutations that can be transmitted to the progeny, but also to ensure correct chromosome segregation during meiosis, the process that forms haploid gametes from diploid germ cells. Accurate chromosome segregation during meiosis requires the deliberate formation of DSBs and their repair as inter-homolog crossover events, which provide the basis for chiasmata, the physical attachments between homologous chromosomes that promote their correct orientation on the first meiotic spindle [20]. Chiasma formation can be easily assessed in diakinesis oocytes (stage preceding metaphase I): wild-type oocytes display 6 pairs of homologous chromosomes attached by chiasmata, while oocytes of crossover-deficient mutants display 12 unattached chromosomes. Oocytes from wild-type controls and *app-1* mutants displayed 6 pairs of attached chromosomes, demonstrating that APP-1 is not required for chiasma formation (Fig 1C). Consistent with this, homologue pairing and synaptonemal complex assembly, two events required for inter-homolog crossover formation, remain intact in *app-1* mutants (S1A and S1B Fig).

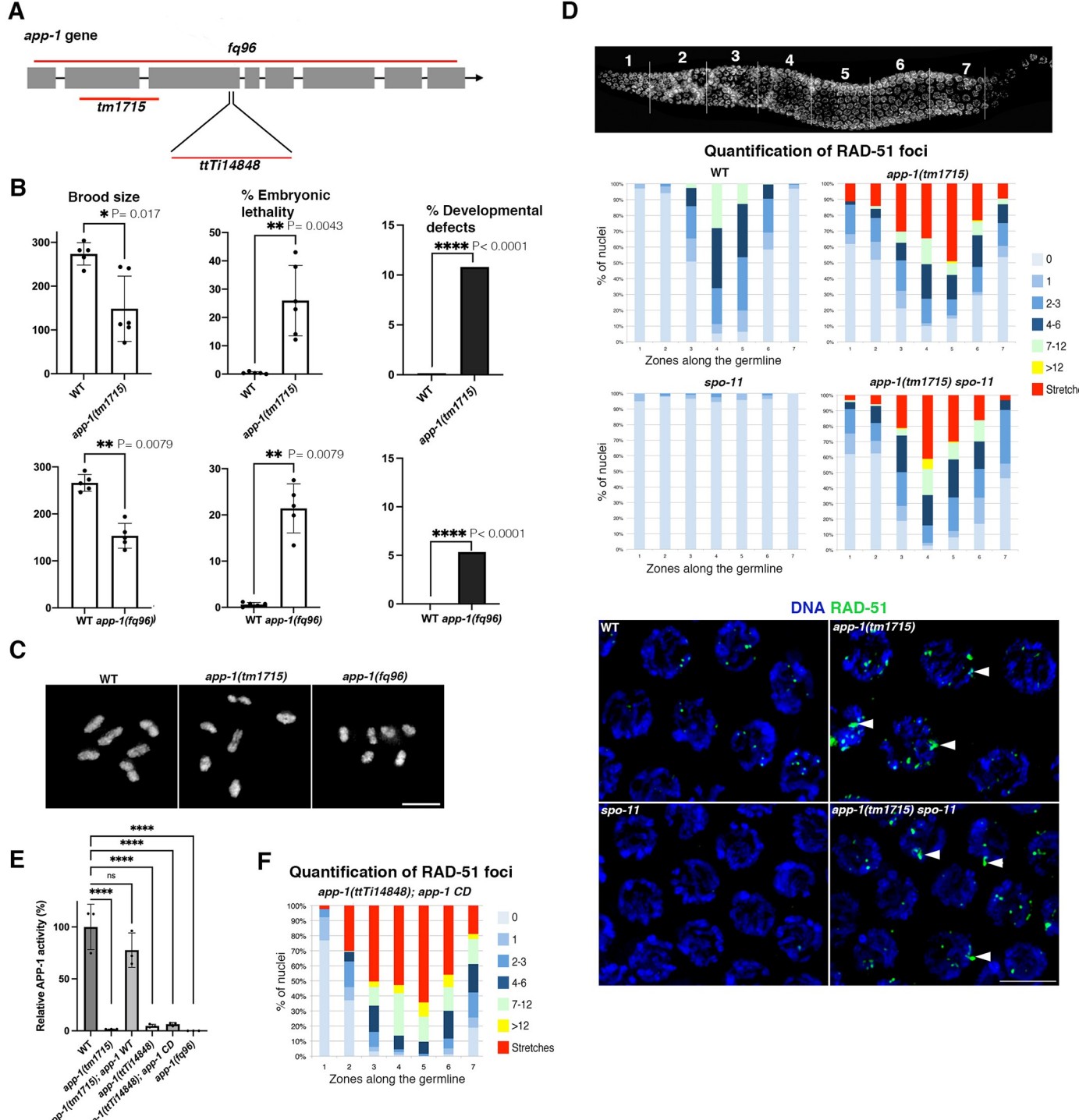

**Fig 1. Loss of APP-1 causes lethality and accumulation of SPO-11-independent DSBs.** (**A**) Schematic representation of the *app-1* locus indicating the regions deleted by the *tm1715* and *fq96* deletion alleles, and the site of insertion in the *ttTi14848* allele. (**B**) Brood size, embryonic lethality, and developmental defects observed among the progeny of homozygous *app-1(tm1715)* and *app-1(fq96)* mutants. Total numbers of embryos scored: 1369 (WT control top row graphs), 891 (*app-1(tm1715)*), 1330 (WT control bottom row graphs), and 767 (*app-1(fq96)*). In graphs for brood size and embryonic lethality circles indicate values from progeny of individual worms, bar indicates mean, error bars show standard deviation, and statistical analysis was calculated using two-tailed nonparametric Mann-Whitney test. % of developmental defects were measured by counting worms with abnormal morphology among the total hatched embryos from each genotype and statistical analysis was calculated using two-sided Chi square test. (**C**) Diakinesis oocytes demonstrating normal chiasma formation in *app-1* mutants. (**D**) Graphs display the regions along the germ line (X axis) as indicated in the DAPI-stained germ line and the percentage of nuclei with a given number of RAD-51 foci (Y axis) as indicated in the color key. RAD-51 foci

accumulate in all germline regions of *app-1(tm1715)* and *app-1(tm1715); spo-11* mutants. Number of nuclei analysed per genotype and zone: WT (143, 265, 140, 156, 127, 114, 96), *app-1(tm1715)* (97, 106, 99, 110, 149, 95, 84), *spo-11* (119, 195,118, 127, 118, 84, 70), *app-1(tm1715); spo-11* (158, 173, 165, 107, 147, 124, 95). (**E**) Enzymatic assay monitoring proteolytic degradation of the APP-1 substrate Lys(εDNP)-Pro-Pro-Amp by soluble protein extracts prepared from an equal weight of wild-type (WT) controls and different *app-1* mutants. APP activity is presented relative to the activity of WT N2, bars indicate mean of three measurements, circles show individual measurements, and errors bars indicate standard deviation. One-way ANOVA test shows that all *app-1* mutants are significantly different from WT controls (**** P<0.0001), while expression of the transgene encoding wild-type APP-1 *[app-1$^{WT}$]* rescues the catalytic activity in *app-1(tm1715)* mutants (P = 0.21 WT vs *app-1 (tm1715) app-1 WT*). No increase in fluorescence was observed with enzyme from *app-1(fq96)*. (**F**) Quantification of RAD-51 foci in worms homozygous for the *app-1 (ttTi14848)* allele and for a transgene expressing a catalytically dead version of APP-1 (carrying the H392A and H496A mutations), note high accumulation of RAD-51 similar to *app-1(tm1715)* and *app-1(fq96)* mutants (see panel D and S1D Fig). Number of nuclei analysed per zone: 130, 159, 161, 110, 126, 74, 90. Scale bar = 5 μm in all panels. See S1 Table for underlaying numerical data of graphs.

Given that the embryonic lethality of *app-1* mutants cannot be explained by a defect in chiasma formation, we investigated if the repair of meiotic DSBs progressed normally. Meiotic DSBs are formed by the SPO-11 protein [21,22] and following their initial processing the resulting ssDNA ends are bound by the recombinase RAD-51, which promotes strand invasion of a homologous template. To assess whether the induction and resolution of recombination intermediates occurred normally in *app-1* mutants we analysed RAD-51 dynamics in intact germ lines. *app-1* mutant germ lines displayed increased RAD-51 foci in both meiotic and mitotically-proliferating nuclei (Figs 1D and S1C and S1D). Notably, germ lines of *app-1* mutants displayed large aggregates of RAD-51 signals (stretches) that were not observed in wild-type controls. The presence of RAD-51 foci in undifferentiated germ cells suggested that some RAD-51 intermediates present in meiotic nuclei might originate from SPO-11-independent DNA damage. Indeed, germ lines of *app-1(tm1715); spo-11* and *app-1(fq96); spo-11* double mutants demonstrated extensive accumulation of RAD-51 foci in mitotic and meiotic nuclei (Figs 1D and S1C). This suggests that a high proportion of RAD-51 foci observed in meiotic nuclei of *app-1* mutants are generated independently of SPO-11 rather than by a defect in the repair of SPO-11-dependent DSBs.

Importantly, the accumulation of RAD-51 foci and embryonic lethality of *app-1(tm1715)* mutants is rescued by a single-copy transgene that expresses wild type APP-1 using the regulatory elements of the *app-1* gene (S1D Fig). These findings confirm that APP-1 is required for normal viability and to prevent the accumulation of SPO-11-independent DNA damage in the germ line.

## The catalytic activity of APP-1 is required to prevent DNA damage accumulation

Since aminopeptidase P has not been previously linked with maintenance of genome stability, we sought to determine if lack of the canonical activity of APP-1 caused the defects observed in *app-1* mutant germ lines. APP-1 belongs to a family of metalloproteases that use an activated water molecule coordinated to a divalent metal ion as the catalytic nucleophile for peptide bond hydrolysis. Mutations of metal ligand residues within the catalytic core of *E. coli* aminopeptidase P inhibit its enzymatic activity without disrupting the overall structure of the protein [23]. Given the structural conservation between *C. elegans* and *E. coli* APP-1 [24], we mutated two conserved histidines (H392A and H496A) known to be required for the catalytic activity of *E. coli* APP1 [23]. The resulting *app-1$^{H392A\ H496A}$* transgene (called *app-1CD* hereafter) was tested for its ability to rescue the *app-1* mutant *app-1(ttTi14848)* that carries a transposon insertion on the third exon of *app-1* (Fig 1A) and also accumulates SPO-11-independent RAD-51 foci (S1E Fig). We then measured cleavage activity towards the APP-1 substrate Lys (εDNP)-Pro-Pro-Amp in whole-worm extracts from wild type controls, *app-1(tm1715)*, *app-1 (fq96)*, *app-1(ttTi14848)*, worms expressing wild-type *app-1* transgene in the *app-1(tm1715)* background, and worms expressing the *app-1CD* transgene in the *app-1(ttTi14848)*

background. While extracts from worms expressing the wild-type *app-1* transgene cleaved the APP-1 substrate with similar efficiency to wild-type controls, the activity of worms expressing *app-1CD* was reduced to levels comparable to *app-1(tm1715)*, *app-1(ttTi14848)*, and *app-1 (fq96)* mutants (Fig 1E), confirming that APP-1$^{H392A\ H496A}$ is indeed a catalytic dead version of APP-1. Importantly, germ lines of *app-1CD* mutants showed extensive accumulation of RAD-51 foci (Fig 1F). Thus, the canonical catalytic activity of APP-1 is required to prevent accumulation of DNA damage in germline nuclei.

### Proliferating germ cells display DNA replication defects in *app-1* mutants

Each adult *C. elegans* germ line contains a compartment of mitotically proliferating germ cells that enter meiosis as they move away from the Notch signalling niche created by the distal tip cell that caps the germ line [25]. Proliferating germ cells display an average cell cycle length of 6–8 hours under unchallenged conditions [26], but undergo cell cycle arrest manifested by enlargement of affected nuclei when replication fails or DSBs are exogenously induced [27]. DAPI staining of *app-1(tm1715)* and *app-1(fq96)* mutant germ lines demonstrated the presence of enlarged nuclei in the mitotic compartment of the germ line, suggesting the presence of arrested nuclei (Fig 2A). Moreover, both *app-1(tm1715)* and *app-1(fq96)* mutants displayed a reduced number of nuclei in the mitotic compartment of the germ line compared to wild-type controls (Fig 2B), suggesting that APP-1 maintains the proliferation of undifferentiated germ cells. In addition, proliferating germ cells of *app-1(tm1715)* mutants displayed increased numbers of replication protein A (RPA-1) foci, consistent with the presence of ssDNA that could arise due to defects during DNA replication (S2A Fig).

To investigate if cell cycle progression is altered in undifferentiated germ cells of *app-1* mutants, we fed young adult worms with EdU labelled bacteria, which allows labelling of DNA undergoing replication in the germ line [28]. Wild-type germ lines dissected following 14 hours of EdU incorporation demonstrated labelling of nearly all nuclei in the mitotic compartment, consistent with nuclei undergoing S-phase at least once over 14 hours (Fig 2C). In contrast, *app-1(tm1715)* mutant germ lines dissected after 14 hours of EdU incorporation contained an average of 15% of nuclei displaying very weak or no EdU labelling (Fig 2C) revealing a population of nuclei that had failed to undergo DNA replication within 14 hours. We further tested if cell cycle progression was compromised in germ cells of *app-1* mutants by examining the population of nuclei undergoing S-phase in a 3-hour time window. Direct injection of labelled nucleotides into the germ line 3 hours before dissection and fixation demonstrated that a larger proportion of mitotically-proliferating nuclei failed to incorporate labelled nucleotides over this time window in *app-1* mutants compared to controls (Fig 2D), consistent with APP-1 promoting normal replication rates. These results demonstrate a defect in germ cell proliferation and suggest that DNA replication is compromised in *app-1* mutants.

### *app-1* mutants show altered response to HU-mediated inhibition of DNA replication, but react normally to irradiation-induced DSBs

Having observed that nuclei in the mitotic compartment of *app-1* mutant germ lines display proliferation defects and accumulation of DNA damage, we investigated how they responded to genotoxic agents known to interfere with DNA replication. We exposed *app-1* mutants to different concentrations of hydroxyurea (HU), a ribonucleotide reductase inhibitor that causes replication fork stalling by depleting deoxynucleotide triphosphate pools [29]. Exposure of wild-type worms to HU causes cell cycle arrest evidenced by enlarged nuclei, a reduction in the number of mitotically proliferating nuclei, and loading of RAD-51 to ssDNA associated with stalled replication forks [30]. In agreement with this, the number of nuclei in the mitotic

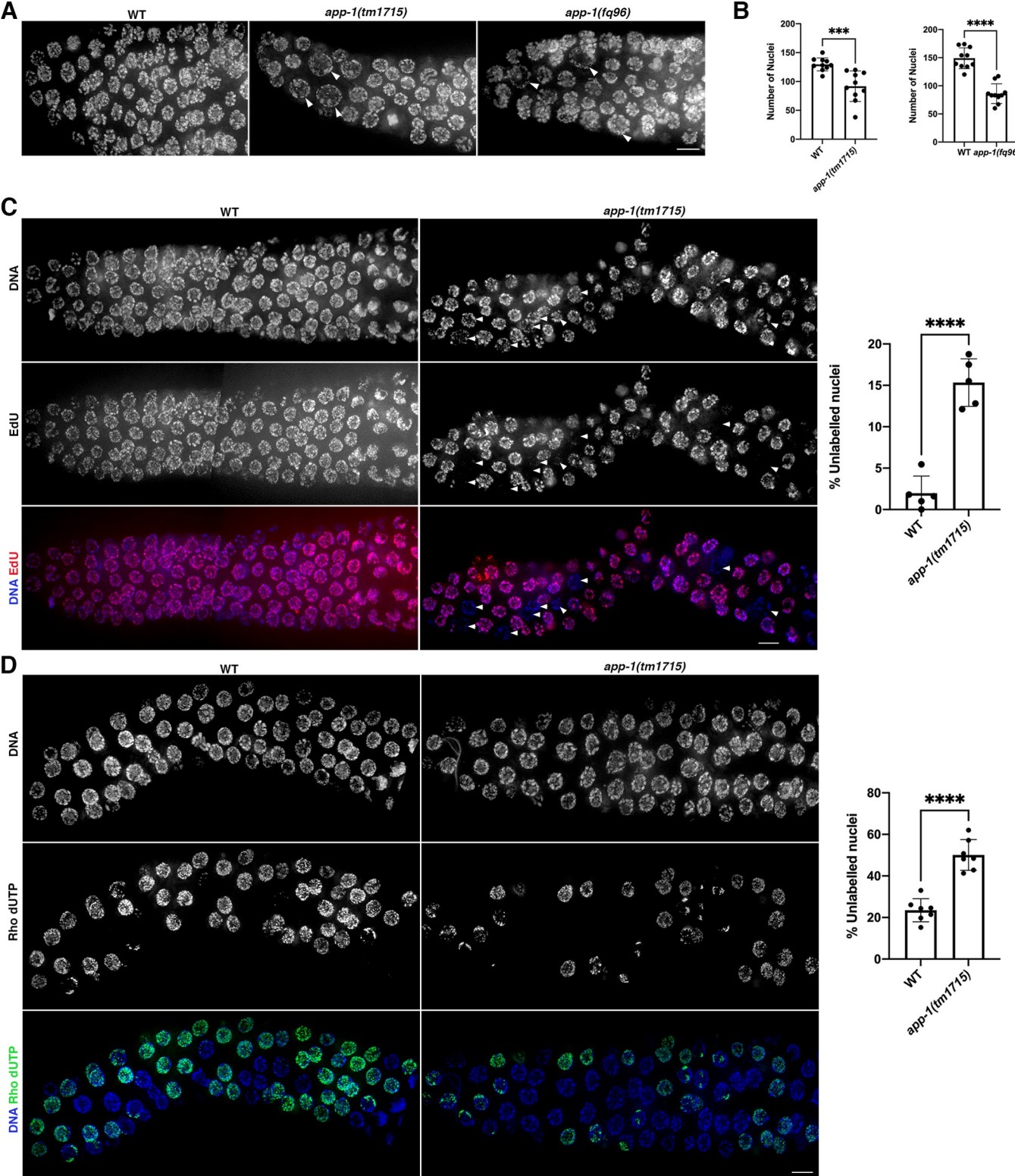

**Fig 2. *app-1* mutant germ lines display DNA replication defects.** All images show projections of the mitotic compartment of the germ line containing mitotically-proliferating germ cells. **(A)** DAPI staining reveals the presence of enlarged nuclei (arrow heads) and an overall reduction in the number of nuclei in *app-1* mutants. **(B)** Graphs show quantification of the number of mitotic nuclei, 10 germ lines were scored per genotype and the total number of nuclei analysed was: 1296 (WT for *app-1(tm1715)*), 918 (*app-1(tm1715)*), 1490 (WT for *app-1(fq96)*), 860 (*app-1(fq96)*). **(C)** Germ lines dissected, fixed, and stained

after 14 hours of feeding on EdU labelled bacteria, note the presence of unlabelled nuclei (arrowheads) in *app-1* mutants. Graph shows quantification of the % of unlabelled nuclei in 5 germ lines per genotype and the total number of nuclei analysed per genotype was: 346 (WT) and 287 (*app-1(tm1715)*). **(D)** Germ lines dissected, fixed, and stained 3 hours after Rho dUTP was injected into the germ line, note increased presence of unlabelled nuclei in *app-1* mutants. Graph shows quantification of the % of unlabelled nuclei in 7 germ lines per genotype and the total number of nuclei analysed per genotype was: 681 (WT) and 571 (*app-1(tm1715)*). In all graphs dots correspond to values of individual germ lines, columns indicate mean value, error bars show standard deviation, and statistical significance was calculated by unpaired t test (**** indicates P value < 0.0001 and *** indicates P value <0.001). Scale bar = 5 μm in all panels. See S1 Table for underlaying numerical data of graphs.

compartment of wild-type germ lines was reduced at all tested doses of HU (5 mM to 40 mM) with the effect of HU peaking at 10 mM (Fig 3A). In contrast, in *app-1(tm1715)* mutant germ lines the appearance of enlarged nuclei and a clear reduction in the overall number of proliferating nuclei was only observed at the highest doses of HU (25 mM and 40 mM), while low HU concentrations had little effect on the number of nuclei (Fig 3A). Similarly, while wild-type germ lines treated with 10 mM HU displayed extensive RAD-51 foci accumulation (80% of nuclei with more than 10 RAD-51 foci), in *app-1* mutants significant accumulation of nuclei with high numbers of RAD-51 foci was only observed after treatment with 40 mM HU and even at this point about 30% of nuclei displayed no RAD-51 foci (Fig 3B). Interestingly, germ lines of wild-type worms treated with 5 mM HU displayed numbers of RAD-51 foci very similar to those seen in untreated *app-1* mutant germ lines, suggesting that treatment of wild-type worms with low doses of HU induces similar type and levels of DNA damage to that resulting from the lack of APP-1. Moreover, inhibition of replication with HU causes elevated levels of topoisomerase II, an enzyme that normally removes supercoiling during DNA replication, in human cells [31] and we observed that *app-1* mutant germ lines also display elevated levels of topoisomerase II (S2B Fig).

To clarify if the delayed appearance of HU-induced RAD-51 foci in *app-1* mutant germ lines reflected a role for APP-1 in promoting RAD-51 loading in mitotically-proliferating nuclei, we monitored RAD-51 focus formation following induction of DSBs by gamma irradiation. We exposed worms to 10 Gy of irradiation and monitored RAD-51 focus formation at 10 and 60 minutes post irradiation, as these conditions successfully identified factors that promote RAD-51 loading to irradiation-induced DSBs in germline nuclei [32]. We observed efficient RAD-51 focus formation in proliferating germ cells of both wild-type controls and *app-1* mutants 10 minutes after irradiation (Fig 3C). Thus, APP-1 is not required for RAD-51 loading at irradiation-induced DSBs. Moreover, we also observed a similar increase in embryonic lethality among the progeny of *app-1* mutants and wild-type worms exposed to increasing doses of gamma irradiation (S3 Fig), confirming that *app-1* mutants are competent in responding to irradiation-induced DNA damage, unlike *brc-1* (BRCA1) mutants used as a negative control.

As the main effect of HU is exerted in cells undergoing DNA replication and *app-1* mutants remain fully competent for RAD-51 loading to irradiation-induced DSBs, our results are consistent with the presence of replication stress in *app-1* mutants.

## *app-1* mutants display synthetic lethality with PARPs

Poly(ADP-ribose) polymerases (PARPs) are rapidly recruited to sites of DNA breaks, where their activation influences signalling and repair of DNA damage [33]. In mammalian cells, PARPs promote survival in the presence of replication stress by promoting the detection and remodelling of stalled replication forks [34]. Given that our data suggests that removal of APP-1 causes replication stress, we investigated if PARPs contribute to DNA repair and viability in *app-1* mutants. Worms carry two PARP homologues (*parp-1* and *parp-*2) so we created double and triple mutant combinations between *parp-1/2* and the *app-1(tm1715)* allele to investigate a

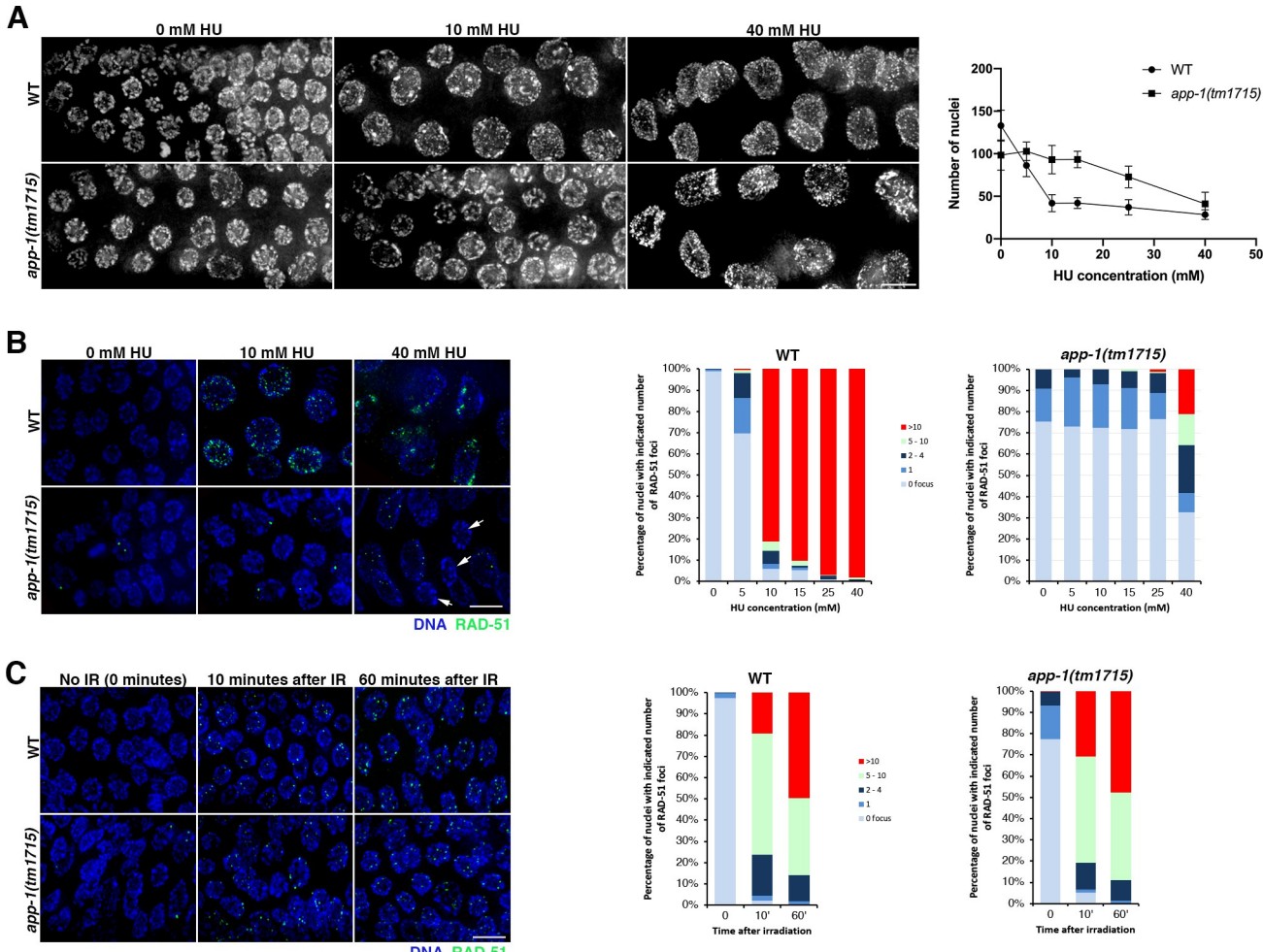

**Fig 3. Effect of exogenous genotoxic stress on *app-1* mutants.** All images show projections of the mitotic compartment of the germ line containing mitotically-proliferating germ cells. **(A)** DAPI staining of dissected germ lines following exposure to the indicated doses of HU for 16 hours. Note that appearance of enlarged nuclei and overall reduction in the number of nuclei requires higher doses of HU in *app-1* mutants. In the graph, individual data points represent the mean and error bars standard deviation, the number of nuclei and germ lines (in brackets) analysed per condition (0 mM HU, 5 mM HU, 10 mM HU, 15 mM HU, 25 mM HU, and 40 mM HU) were: WT: 1330 (10), 863 (10), 419 (10), 420 (10), 370 (10), and 227 (8); *app-1(tm1715)*: 984 (10), 1029 (10), 930 (10), 933 (10), 727 (10), and 410 (10). **(B)** RAD-51 loading following exposure to the indicated doses of HU for 16 hours, note that significant accumulation of RAD-51 foci in *app-1* mutants is only observed with 40 mM HU. Arrows in the *app-1* panel treated with 40 mM HU point to nuclei displaying no RAD-51 foci. Graphs show quantification of RAD-51 foci in mitotically-proliferating germ cells, the number of nuclei and germ lines (in brackets) analysed per condition (0 mM HU, 5 mM HU, 10 mM HU, 15 mM HU, 25 mM HU, 40 mM HU were: WT: 605 (5), 355 (5), 238 (5), 219 (5), 193 (5), and 216 (6); *app-1(tm1715)*: 359 (5), 356 (5), 364 (5), 423 (5), 350 (6), and 237 (6). **(C)** RAD-51 loading before (time 0), 10 minutes and 60 minutes after exposure to 10 Gy of γ irradiation, note similar levels of RAD-51 foci in WT and *app-1* mutant germ lines. 5–6 germ lines were quantified per genotype and condition. Total number of nuclei: 638 (WT at 0), 645 (WT at 10'), 535 (WT at 60'), 488 (*app-1* at 0). 473 (*app-1* at 10'), and 400 (*app-1* at 60'). Scale bar = 5 μm in all panels. See S1 Table for underlaying numerical data of graphs.

potential contribution of PARPs to the viability of *app-1* mutants. *parp-1/2* single and double mutants display near-normal levels of embryonic lethality and developmental abnormalities among their progeny (Fig 4A and 4B), revealing that PARPs are not essential for viability under unchallenged conditions. While *parp-1*, *app-1* and *app-1; parp-2* double mutants displayed defects similar to *app-1* single mutants, simultaneous removal of PARP-1/2 from *app-1* mutants induced high levels of embryonic lethality and developmental defects among the viable progeny (Fig 4A and 4B). The synthetic lethality between *app-1* and *parp-1/2* suggests that PARPs play an important role in repairing DNA damage caused by the absence of APP-1.

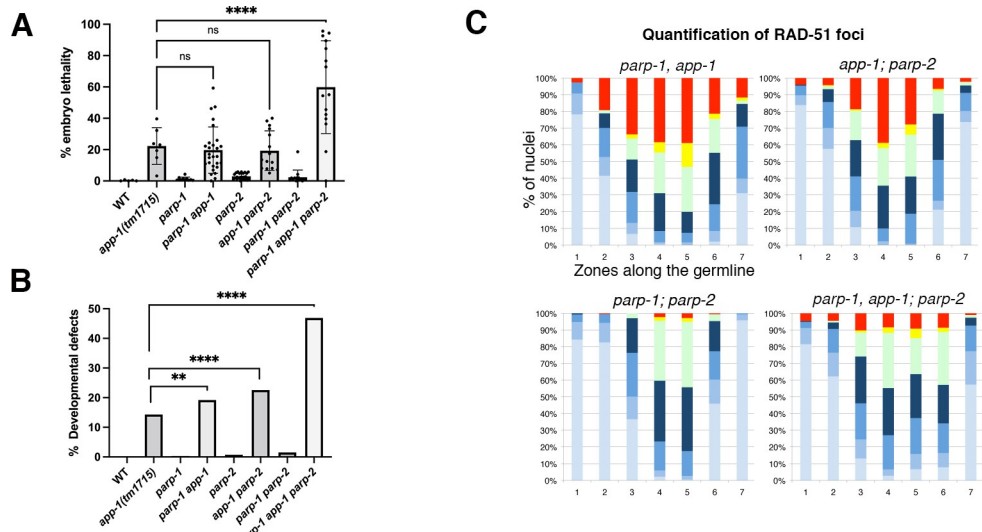

**Fig 4. poly(ADP-ribosyl)ation contributes to DNA repair in *app-1* mutants. (A-B)** Removal of PARP-1 and PARP-2 from *app-1(tm1715)* mutants increases embryonic lethality and the incidence of developmental defects amongst viable progeny. Number of embryos scored per genotype: 1275 (WT), 941 (*app-1*), 2361 (*parp-1*), 3449 (*parp-1, app-1*), 5040 (*parp-2*), 1872 (*app-1; parp-2*), 2209 (*parp-1; parp-2*), and 1401 (*parp-1, app-1; parp-2*). In (A) dots indicate % of embryonic lethality among the progeny of individual worms, bars indicate mean value of % embryonic lethality from all scored worms, and error bars indicate standard deviation. Statistics were calculated using a one-way ANOVA test, P values = 0.998 (*app-1* vs *parp-1, app1* and *app-1* vs *app-1; parp-2*); <0.0001 (*app-1* vs *parp-1, app-1; parp-2*). In (B) % of developmental defects were measured by counting worms with abnormal morphology among the total hatched embryos from each genotype, statistical analysis was performed using two-sided Chi square test (**** indicates P value < 0.0001 and ** indicates P = 0.002). **(C)** Quantification of RAD-51 foci in germ lines of indicated genotypes, graphs display the regions along the germ line (X axis) and the percentage of nuclei with a given number of RAD-51 foci (Y axis) as indicated in color key. Note that removing PARP-1 and PARP-2 reduces the accumulation of RAD-51 aggregates in germ lines of *app-1(tm1715)* mutants, but RAD-51 foci still accumulate to high levels in germ lines of *app-1; parp-1* and *app-1; parp-2* double mutants. The number of nuclei analysed per genotype and zone were: *parp-1, app-1(tm1715)* (152, 161, 119, 154, 175, 94, 103), *app-1; parp-2* (155, 168, 102, 129, 112, 94, 137), *parp-1; parp-2* (287, 265, 237, 184, 181, 194, 169), and *parp-1, app-1(tm1715); parp-2* (195, 183, 167, 181, 209, 152, 190). See S1 Table for underlaying numerical data of graphs.

In human tissue culture cells, the joint activity of PARP1/2 promotes loading and stabilization of RAD51 at damaged replication forks [35], prompting us to investigate whether PARP-1/2 contribute to RAD-51 loading in germ lines of wild-type worms and *app-1* mutants. We observed no obvious defects in the loading and removal of RAD-51 in germ lines of *parp-1/2* single and double mutants (Figs 4C and S4A–S4C), demonstrating that PARPs are not required for RAD-51 loading at SPO-11-induced DSBs. In contrast, removing PARPs from *app-1* mutants caused a reduction in the number of nuclei displaying RAD-51 stretches compared with *parp-1, app-1* and *app-1; parp-2* double mutants (Figs 4C and S4C), suggesting that also in nematodes PARP-1/2 might exert a regulatory function on RAD-51 loading and stabilization upon replicative stress. These observations reveal that PARPs contribute to the viability of worms lacking APP-1, likely by promoting early steps in the repair of replication-associated DNA damage.

## MRE-11 and COM-1 (CtIP) promote RAD-51 accumulation in meiotic nuclei of *app-1* mutants

Our observations so far reveal that lack of APP-1 causes DNA replication defects and formation of SPO-11-independent DNA damage in the mitotic and meiotic regions of the germ line.

We also noted that in *app-1; spo-11* double mutants the accumulation of RAD-51 foci increases as nuclei enter meiosis (zones 2–3 in RAD-51 foci quantification of graphs in Figs 1D and S1D and S1F), suggesting that some aspect of the early meiotic program could contribute to the loading of RAD-51 at DNA damage sites originated during DNA replication in *app-1* mutants. We hypothesised that factors that operate downstream of SPO-11 during meiotic recombination may also contribute to RAD-51 loading in the absence of APP-1. Thus, we investigated if the conserved MRE-11 (Mre11) and COM-1 (CtIP) nucleases, which are required for the resection of SPO-11-induced DSBs [36–38] promote RAD-51 loading in *app-1* mutant germ lines. As expected, germ lines of *mre-11* and *com-1* single mutants lacked SPO-11-dependent RAD-51 foci and instead displayed low levels of RAD-51 foci in about 15–20% of nuclei throughout the germ line (Fig 5A and 5B). The percentage of nuclei displaying RAD-51 foci in germ lines of *app-1; mre-11* and *app-1; com-1* double mutants was slightly higher than that observed in *com-1* or *mre-11* single mutants (Fig 5A and 5B). However, the number of RAD-51 foci in germ lines of *app-1; mre-1* and *app-1; com-1* double mutants was severely reduced compared with both *app-1* single and *app-1*; *spo-11* double mutants (see Figs 1D and S1C). Moreover, *app-1; mre-11* and *app-1*; *com-1* double mutants also lack the clear increase in RAD-51 foci observed at meiotic entrance in germ lines of *app-1; spo-11* double mutants. These observations confirm that MRE-11 and COM-1 promote the early processing of SPO-11-independent DNA damage in *app-1* mutants, suggesting that homologous recombination plays a major role in the repair of DNA lesions accumulated in the absence of APP-1.

### *app-1* mutants repair SPO-11-independent DNA damage as crossovers using the canonical meiotic recombination pathway

In *spo-11* mutants homologue pairing and synapsis remain intact and the introduction of exogenous DSBs by gamma irradiation is sufficient to restore chiasma formation [21]. Given that pairing and synapsis also remain intact in *app-1* mutants (S1A Fig), we asked whether the SPO-11-independent DNA damage that accumulates in the absence of APP-1 could be converted into chiasmata by imaging diakinesis stage oocytes of *app-1; spo-11* double mutants. As expected, oocytes of wild-type controls and *app-1* mutants contained 6 DAPI-stained bodies (6 pairs of homologous chromosomes attached by chiasmata) and oocytes of *spo-11* mutants displayed 12 DAPI-stained bodies (12 unattached chromosomes) (Fig 6A). Strikingly, oocytes of *app-1(tm1715); spo-11* and *app-1*(*fq96); spo-11* displayed an average of ~7.2 DAPI-stained bodies (Fig 6A), revealing extensive chiasma formation and suggesting that SPO-11-independent DNA damage present in *app-1* mutants results in DSBs that can be repaired as inter-homologue crossover events.

To further test this possibility, we assessed chiasma formation in oocytes of mutants lacking APP-1 and either the SC component SYP-2 or the MutS component MSH-5, two factors required for the formation of inter-homologue crossover events during meiosis [39, 40]. *app-1; syp-2* double mutants oocytes displayed an average of 11.4 DAPI-stained bodies, while *app-1; msh-5* double mutants displayed an average 10.5 (Fig 6A), confirming that most chiasmata observed in *app-1; spo-11* oocytes require SC assembly and MSH-5. However, *app-1; msh-5* double mutant oocytes displayed a statistically significant reduction in the number of DAPI-stained bodies compared to *msh-5* single mutants, revealing that some of the chromosomal attachments that we scored as chiasmata in these oocytes were formed in an MSH-5-independent fashion. As virtually all inter-homologue crossovers are MSH-5-dependent in wild-type worms [40], this suggested the involvement of additional DNA damage repair mechanisms. In agreement with this possibility, FISH analysis of diakinesis oocytes confirmed the presence of non-homologous chromosome fusions in oocytes of *app-1; spo-11* double mutants (S5 Fig),

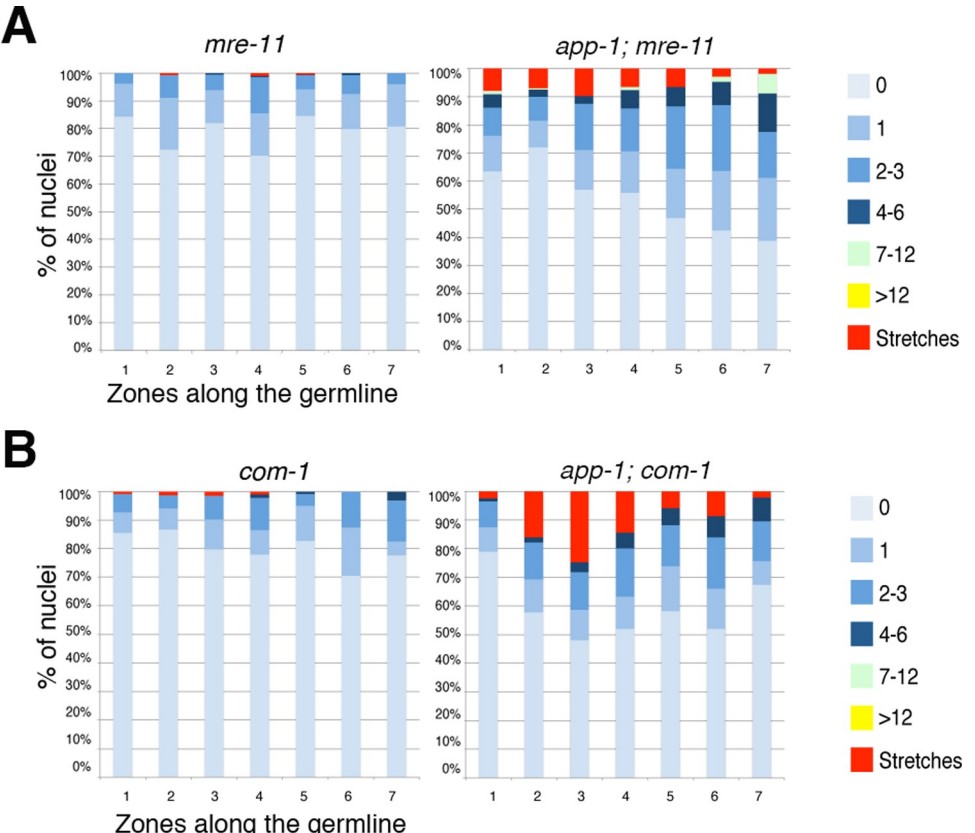

**Fig 5. MRE-11 and COM-1 promote RAD-51 accumulation in *app-1* mutant germ lines.** Quantification of RAD-51 foci in germ lines of indicated genotypes, graphs display the regions along the germ line (X axis) and the percentage of nuclei with a given number of RAD-51 foci (Y axis) as indicated in color key. Note that removing MRE-11 or COM-1 from *app-1* mutants causes a reduction in the number of RAD-51 observed in meiotic nuclei compared to *app-1* single and *app-1; spo-11* double mutants (see Fig 1D). Number of nuclei analysed per genotype and zone: *mre-11* (211, 178, 245, 226, 241, 175, 98), *app-1(tm1715); mre-11* (101, 199, 193, 170, 171, 146, 103), *com-1* (125, 150, 183, 177, 122, 126, 63), and *app-1(tm1715); com-1* (167, 177, 121, 125, 153, 150, 95). See S1 Table for underlaying numerical data of graphs.

suggesting that some SPO-11-independent DSBs present in *app-1* mutants may be repaired by pathways other than homologous recombination such as non-homologous end joining.

To obtain direct evidence for the formation of canonical crossover events in germ lines of *app-1; spo-11* double mutants we used CRISPR-tagged versions of MSH-5 and COSA-1 to visualise these pro-crossover factors [41,42]. MSH-5 and COSA-1 colocalise to recombination foci in late pachytene nuclei, forming a single focus per pair of homologous chromosomes that reveals the position of the single crossover event formed per homologue pair during oocyte meiosis in worms [43]. While MSH-5 and COSA-1 foci were largely lacking from germ lines of *spo-11* mutants, *app-1; spo-11* double mutants displayed MSH-5 and COSA-1 foci from early pachytene (Fig 6B). Similar to the situation in wild-type controls and *app-1* single mutants, most MSH-5 and COSA-1 foci detected in late pachytene nuclei of *app-1; spo-11* double mutants localise together, suggesting the formation of functional crossover-recombination sites (Fig 6B). These observations provide evidence that a large proportion of the SPO-11-independent DNA damage caused by the absence of APP-1 includes DSBs that are repaired by the canonical crossover pathway, leading to the formation of chiasmata.

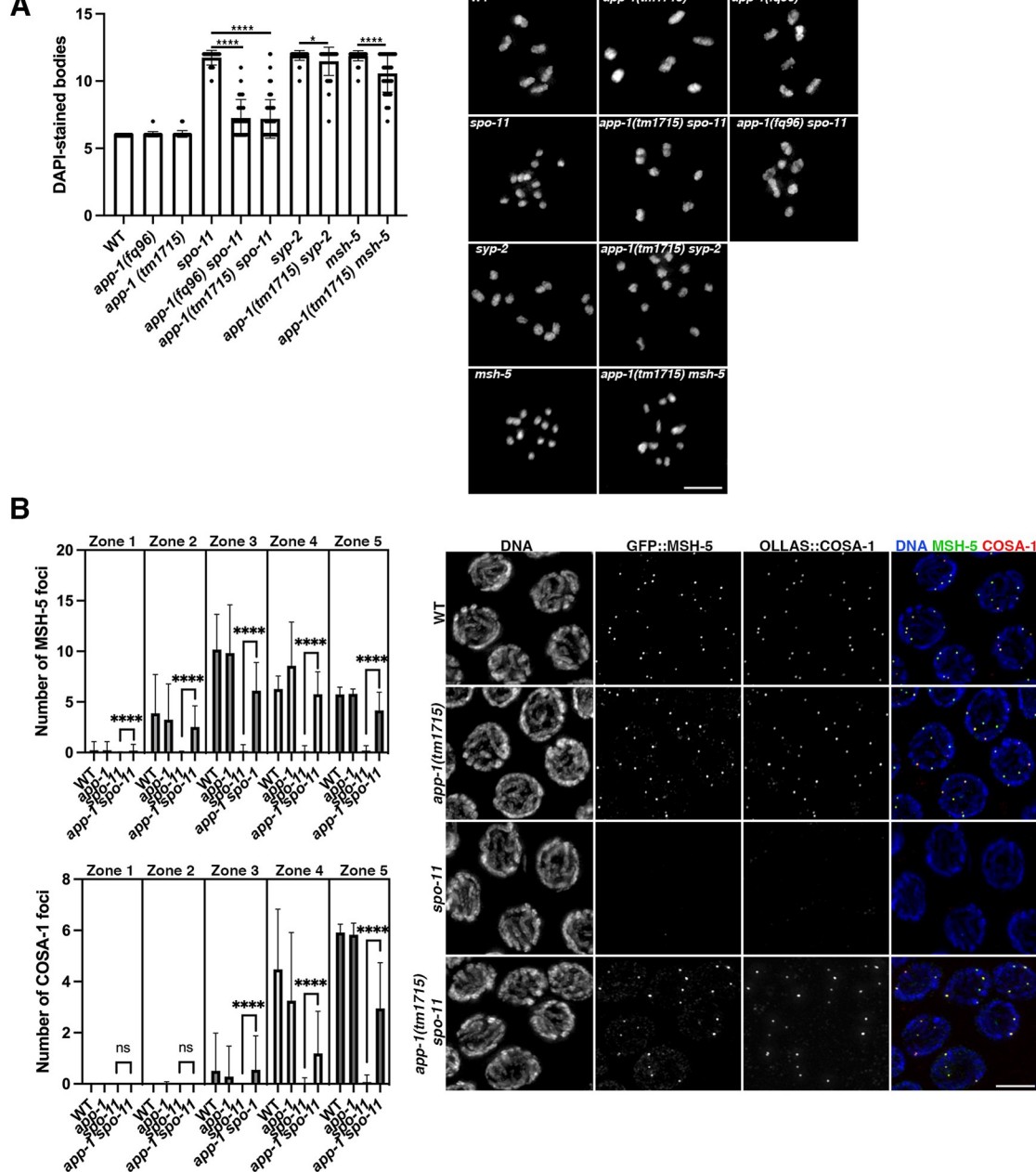

**Fig 6. The crossover pathway repairs DSBs in *app-1* mutants.** (**A**) Quantification of the number of DAPI-stained bodies in diakinesis oocytes of indicated genotypes. Columns indicate mean value and error bars show standard deviation. Statistical significance was calculated by two tailed Mann Whitney U test (**** indicates P value < 0.0001 and * indicates P value = 0.01420). Number of oocytes analysed: WT (25), *app-1(fq96)* (25), *app-1(tm1715)* (61), *spo-11* (23), *app-1(fq96) spo-11* (27), *app-1(tm1715) spo-11* (85), *syp-2* (43), *app-1(tm1715) syp-2* (60), *msh-5* (53), *app-1 (tm1715) msh-5* (58). Images show projections of diakinesis oocytes of indicated genotype stained with DAPI. (**B**) Quantification of MSH-5 and COSA-1 foci in germ lines of indicated genotypes. Graphs show zones along the germ line from transition zone (1) to late pachytene (5) on the X axis and the number of foci per nucleus on the Y axis. Columns indicate mean value and error bars show standard deviation. Statistical significance between numbers of MSH-5 and COSA-1 foci between *spo-11* and *app-1 spo-11* were calculated by a two tailed Mann Whitney U test (**** indicates P value < 0.0001). The number of nuclei analysed per genotype and zone were: WT (236, 230, 186, 101, 70), *app-1(tm1715)* (214, 202, 177, 121, 93), *spo-11* (244, 243, 208, 137, 118), and *app-1(tm1715) spo-11* (310, 218, 198, 156, 122). Images show projections of late pachytene nuclei stained with anti-GFP (MSH-5), anti-OLLAS (COSA-1), and DAPI. Note that COSA-1 and MSH-5 foci are largely absent in *spo-11* mutants, but present in *app-1 spo-11* double mutants. Scale bar = 5 μm in all panels. See S1 Table for underlaying numerical data of graphs.

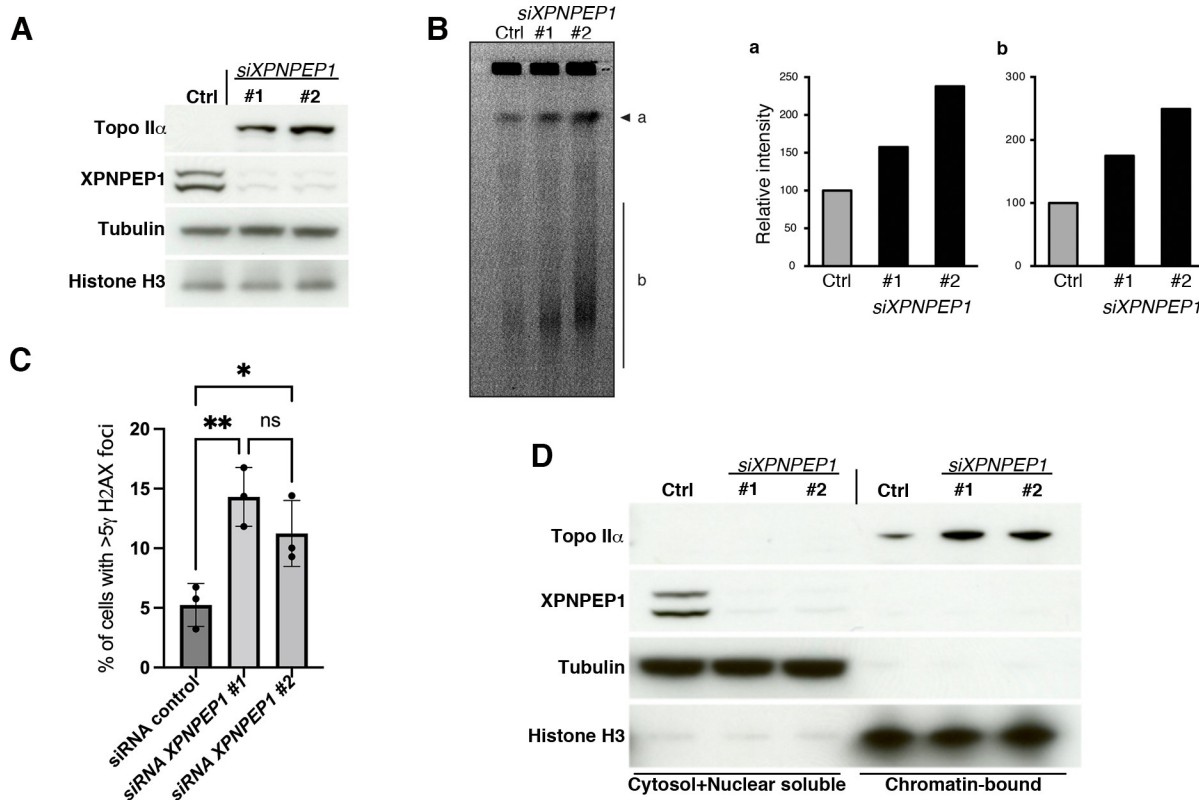

**Fig 7. Knockdown of human XPNPEP1 causes accumulation of DNA damage. (A)** Western blot analysis on total protein extracts showing efficient knock down of *XPNPEP1* by two independent siRNAs that also induce accumulation of Topoisomerase IIα. **(B)** Pulse field gel stained with ethidium bromide in control cells and following knock down of *XPNPEP1*. Note increased intensity of band labelled as "a" and the smear label as "b" corresponding to fragmented DNA. Graphs show quantification of the relative intensity of band "a" and smear "b". **(C)** *XPNPEP1* knockdown causes an increase in the % of cells positive for γH2AX. Dots show % of cells with >5 γH2AX foci from three experiments, bars show mean of the three experiments, and error bars represent standard deviations. Statistical significance was calculated by a one-way ANOVA test, note that differences between siRNA control and the two siRNAs targeting *XPNPEP1* are significant (* indicates P value = 0.048 and ** indicates P value = 0.0082). **(D)** Western blot analysis on fractionated protein extracts from U2OS cells treated with control siRNA and two independent *XPNPEP1* siRNAs showing that Topoisomerase II accumulates on the DNA-bound fraction following *XPNPEP1* knockdown.

### Removal of XPNPEP1 causes DNA damage in human cells

Given that cytosolic aminopeptidase P is highly conserved through evolution [24,44,45] and that mice and flies display embryonic lethality and developmental defects [16–18], we sought to clarify whether the human homologue of *app-1* (*XPNPEP1)* is also involved in regulating genome integrity. We knocked down *XPNPEP1* expression in human U2OS cells using two different siRNAs and confirmed the efficient depletion of XPNPEP1 by both siRNAs (Fig 7A). We then assessed the presence of DNA damage by monitoring DNA fragmentation in pulse field gels and formation of γH2AX foci using immunostaining. Both siRNAs caused an increase in the levels of fragmented DNA and in the percentage of cells displaying γH2AX foci (Fig 7B and 7C). Interestingly, similar to the observations in worms, depletion of XPNPEP1 also caused increased levels of topoisomerase II in U2OS cells (Fig 7A) and fractionation experiments confirmed that this increase occurred in the DNA-bound fraction (Fig 7D). These experiments confirm that knockdown of XPNPEP1 causes accumulation of DNA damage in human cells, suggesting that the role of APP1 in promoting genome integrity is conserved between nematodes and humans.

## Discussion

Genome instability and replicative stress are common features of cancer and precancerous cells as well as aging, underscoring the importance of discovering factors that prevent the onset of these events. We have identified aminopeptidase P as a conserved factor required for genome stability in *C. elegans* and human cells. In the absence of APP1, mitotic and meiotic nuclei accumulate high levels of DSBs, as demonstrated by the striking recovery of crossover intermediates (COSA-1 and MSH-5 foci) and chiasmata in germ lines of worms lacking APP-1 and the SPO-11 nuclease responsible for DSB formation during meiotic recombination. Despite accumulating DSBs, *app-1* mutants appear fully competent in DNA repair by homologous recombination. This is supported by the requirement of resection factors MRE-11 and COM-1 for accumulation of RAD-51 foci in *app-1* mutants, by the ability of *app-1* mutants to repair DSBs induced by ionising radiation, by the formation of normal numbers of crossover foci and chiasmata in germ lines of *app-1* mutants, and by the requirement of MSH-5 and SYP-2 for most chiasmata observed in *app-1; spo-11* double mutants. Instead, defects observed in the mitotic compartment of the germ line, which include an overall reduction in the number of proliferating germ cell nuclei, reduced incorporation of labelled nucleotides, presence of arrested nuclei, delayed response to HU treatment, and increased levels of RPA-1 and RAD-51 foci are consistent with the presence of replication-associated DNA damage in *app-1* mutants. This possibility is further suggested by accumulation of SPO-11-independent RAD-51 foci in mitotic and meiotic nuclei of *app-1* mutant germ lines and by the synthetic lethality observed in mutants lacking APP-1 and both poly(ADP-ribose) polymerases. We propose that cells lacking APP-1 suffer elevated levels of replication stress that are sufficient to temporally overwhelm their repair capability, leading to genome instability.

The requirement of APP-1 in preventing genome instability is clearly demonstrated by the high level of DNA damage that accumulates in *app-1; spo-11* double mutant germ lines. Under normal conditions removal of SPO-11 results in the near complete disappearance of RAD-51 foci, COSA-1 foci, and chiasmata from the germ line, evidencing that SPO-11 is the main source of DSBs and that replication-related DNA damage is mostly absent or efficiently repaired before meiotic entry. Moreover, mutants lacking proteins required for early processing of DSBs during homologous recombination (MRE-11, COM-1, NBS-1, RAD-51) display chromosome aggregation in diakinesis oocytes as DSBs are repaired by NHEJ, however removal of SPO-11 supresses this phenotype [46–49], consistent with low levels of replication-related DNA damage in these mutants. In contrast, *app-1; spo-11* double mutants display levels of RAD-51 foci higher than those of wild-type germ lines. Interestingly, there is a clear increase of RAD-51 foci as nuclei enter meiosis in germ lines of *app-1; spo-11* double mutants, suggesting either that some feature of meiotic chromosome morphogenesis promotes the processing of DNA damage accumulated during the preceding mitotic divisions or that most DNA damage accumulates during meiotic S-phase or early prophase. *app-1; spo-11* double mutants also display COSA-1 foci and chiasmata, demonstrating processing of DSBs by homologous recombination. In fact, we observed that around 60% of diakinesis oocytes in *app-1; spo-11* double mutants have a full complement of chiasmata (6 bivalents); such rescue in chiasma formation in a *spo-11* mutant background has only been previously observed by exposing *spo-11* mutants to gamma irradiation [21].

The level of DSB accumulation in *app-1* mutant germ lines is much higher than that observed in mutants lacking proteases responsible for the removal of DNA protein crosslinks, a type of DNA lesion that can induce replication-related DNA damage [50]. For example, mutants lacking the GCNA-1 protease do not accumulate RAD-51 foci above wild-type levels and *gcna-1; spo-11* double mutants display a very mild increase in the number of RAD-51 foci

and chiasmata compared to *spo-11* mutants [51]. Similarly, we have observed that mutants lacking the DVC-1(Spartan) [52] protease and SPO-11 fail to form chiasmata (S6 Fig). *C. elegans* mutants lacking TOP-3, a component of the STR/BTR complex that participates in homologous recombination, also display high levels of SPO-11-independent DNA damage, but in contrast to *app-1; spo-11* mutants chiasma formation is not rescued in *top-3; spo-11* double mutants [53]. Moreover, chiasma formation is also deficient in *top-3* single mutants, demonstrating that TOP-3, unlike APP-1, is required for meiotic recombination. These phenotypic comparisons with mutants lacking different DNA repair factors highlight the key role of APP-1 as a guardian of genome stability and point to APP-1 acting to prevent the onset of replication-related DNA damage rather than being required for its repair. This possibility is further suggested by the synthetic lethality between *app-1* and *parp-1/2* mutants and by the delayed response (requiring higher doses) of *app-1* mutant germ cells to HU treatment, which specifically targets cells undergoing DNA replication. The importance of identifying factors that protect cells from replication stress is highlighted by recent findings suggesting that DNA-repair-proficient cells can become cancerous by bursts of genomic instability, which can be facilitated by replication and transcription stress [1].

Why does lack of APP-1 induce replication stress? We consider three non-mutually exclusive possibilities to explain how APP-1 could contribute to genome integrity. First, APP-1 could directly modulate the functional properties of proteins involved in DNA metabolism that contain proline residues near their N-terminus. Modification of the N-terminus, including iMet removal and N-terminal acetylation, is emerging as an important mode of regulating protein stability and activity [54] and proline has an important role in this process as a proline residue at position 2 prevents N-terminal acetylation [55]. This process is clearly relevant for proteins involved in chromosome metabolism as N-terminal acetylation of synaptonemal complex component SYP-1 is essential for meiosis in *C. elegans* and mutating residue 2 of SYP-1 from aspartate to proline causes severe meiotic defects [56]. Moreover, the recent identification of an N-terminus proline degron [57] further highlights how the presence of proline at the N-terminus of proteins can be used to regulate protein stability. Interestingly, APP-1 was recently identified as an interactor of PID-5, a protein involved in epigenetic gene silencing in *C. elegans*, suggesting that N-terminal processing of proteins containing proline residues may be important for the deposition of epigenetic marks [58]. Second, aminopeptidases are required to degrade short peptides produced from the proteasome into amino acids. In the absence of APP-1, short proline-containing peptides are expected to accumulate, which could interfere with functionality of the UPS or other metabolic processes. Third, impaired digestion of proline-containing peptides into amino acids is also expected to reduce cellular levels of free proline, which could promote formation of ROS as seen in yeast mutants deficient in proline biosynthesis [59]. A replisome-associated redox sensor has recently been shown to be a key regulator of DNA replication [60], therefore, if lack of APP-1 alters redox status of the cell, this could lead to DNA replication defects such as those observed in *app-1* mutants. Clarifying the mechanisms by which APP-1 promotes genome stability is an important goal for future studies.

Finally, inhibition of proteins that ensure efficient progression of DNA replication leads to genome instability and ultimately cell death, making these proteins key targets of chemotherapy protocols used to treat human malignancies. Our findings that lack of APP-1(XPNPEP1) causes replication-related DNA damage in worms and genome instability in human cells raise the question of whether inhibition of this aminopeptidase could have therapeutic relevance. Future development of XPENPEP1 inhibitors will clarify whether this aminopeptidase could be exploited as a therapeutical target.

## Materials and methods

### C. elegans genetics

The N2 Bristol strain was used as the wild-type control and all worms were kept at 20˚C. The following mutant alleles were used for this study: LG I: *app-1(tm1715)*, *app-1(ttTi14848)*, *app-1(fq96)*, *parp-1(ok988)*; LG II: *parp-2(ok344)*, *top-2(av64[top-2::3XFLAG])*; LG III: *brc-1 (tm1145)*, *brd-1(dw1)*, *com-1(t1626)*, *cosa-1(ddr12[OLLAS::cosa-1])*; LG IV: *fcd-2(tm1298)*, *spo-11(ok79)*, *msh-5(me23)*, *msh-5(ddr22[GFP::msh-5])*; LG V: *mre-11(ok179)*, *dvc-1(ok260)*. Homozygous *app-1* mutants of the different alleles (*tm1715*, *fq96*, *ttTi14848*) that were used for the analysis reported in the manuscript were obtained from homozygous *app-1* mutant mothers. Heterozygous stocks of the different *app-1* alleles were also created for long-term storage of the strains.

Transgenic worms were created by inducing single-copy insertions of the desired transgene following mobilization of a single Mos1 transposon by injection of the Mos transposase [61]. The transgene expressing a wild-type copy of *app-1(fqSi19)* included the 5' UTR upstream of *app-1*, the *app-1* locus and the 3' UTR region downstream of *app-1*. In order to generate a catalytic dead version of APP-1 (APP-1$^{CD}$), the *fqSi19* transgene was modified to encode an APP-1 protein including the H392A and H396A mutations to produce the *fqSi339* transgene. Both transgenes were inserted into the *ttTi5605* locus on chromosome II. Worms bearing the *fqSi19* transgene were crossed into the *app-1(tm1715)* mutant background and those carrying the *fqSi339* transgene (APP-1$^{CD}$) were crossed into the *app-1(ttTi14848)* mutant background. The *app-1(fq96)* allele deleting 2004 bp (out of 2177) of the *app-1* locus was generated by CRISPR using two guide RNAs (ATCCTTGTTCCATTCGGAAC and CAAGTCTCTTCTGATTGA AG) and the following single-stranded DNA oligo as a repair template: AAAAACTCGCAA AATTGAGATCCTTGTTCCATTCGTACCCATACGATGTTCCAGATTACGCTAAGAG GAAATCAATTGGCTCAATCAATACCACGCT.

### Immunostaining and Fluorescence In Situ Hybridization

FISH and immunostaining were performed as in Silva *et al* (2014) [62]. Briefly, immunostaining was performed as follows: germ lines from 18–24 hours post L4 adults were dissected in EGG buffer (118 mM NaCl, 48 mM KCl$_2$, 2mM CaCl$_2$, 2mM MgCl$_2$, 5mM HEPES) containing 0.1% Tween and immediately fixed in 1% paraformaldehyde for 5 minutes. Slides were frozen in liquid nitrogen, then immersed for at least 1 minute in methanol at –20˚C and transferred to PBST (1x PBS, 0.1% Tween). Blocking in 1% BSA in PBST was carried out for 1 hour. Primary antibodies were incubated overnight at room temperature, slides were then washed 3 times for 10 minutes in PBST and secondary antibodies were added and incubated for 2 hours at room temperature. Following 2 washes in PBST, the slides were counterstained with DAPI and mounted using Vectashield. All images were acquired as three-dimensional stacks on a Delta Vision system equipped with an Olympus 1X70 microscope. The following primary antibodies were used: rabbit anti-HTP-1, 1:400, chicken anti-SYP-1, 1:350, rabbit anti-RAD-51, 1:10.000 (SDIX), mouse anti-FLAG, 1:500 (Sigma). Secondary antibodies were used at 1:500 dilution and included: goat anti-rabbit Alexa488-conjugated, goat anti-chicken Alexa555-conjugated and goat anti-mouse Alexa488-conjugated (Life Technologies).

### Monitoring viability and sensitivity to DNA damaging agents in C. elegans

In order to quantify viability levels and developmental abnormalities, L4 worms grown under physiological conditions were individually picked and transferred onto fresh plates every 24 hours. The presence of dead embryos was assessed 24 hours after the mothers had been

removed from each plate and the presence of worms carrying developmental abnormalities in the progeny was scored three days later. For sensitivity to IR, worms were exposed to increasing doses of γ-rays and allowed to lay eggs for 24 hours. Mothers were then removed and dead embryos were scored after one day. To investigate the effect of hydroxyurea (HU) on mitotically-proliferating germ cells L4 worms were picked onto NGM plates containing different concentrations of HU (5 mM, 10 mM, 15 mM, 25 mM and 40 mM) and OP50 *E. coli* as food. Worms were left on the plates for 16 hours before being dissected and fixed for DAPI and antibody staining as described in the immunostaining protocol.

### EdU and Rho-dUTP incorporation

Young adults were fed for the indicated time onto NG plates seeded with the thymine auxotroph bacterial strain MG1693 (*thyA⁻*), which was previously grown in presence of EdU. EdU detection was performed with the Click-iT kit (ThermoFisher) according to the manufacturer's instruction. Rho-dUTP incorporation was carried by direct microinjection into the germ line with a mix containing 25 pmol of tetramethyl-Rhodamine-5-dUTP (Roche). Following injection, worms were picked onto NGM plates seeded with *E. coli* for 3 hours before being dissected and fixed as described in the immunostaining protocol.

### RAD-51 loading following exposure to γ-rays

Worms were exposed to 10 Gy of γ-rays and 10 minutes later the germ lines were dissected, fixed, and processed for anti-RAD-51 staining as described in the immunostaining protocol.

### Aminopeptidase P activity of *C. elegans* extracts

Worms were washed from replicate 5 cm diameter plates in M9 buffer and were harvested after settling under gravity at 4˚C. The worms were then washed 3 times in 10 ml of M9 and stored as a pellet, which was weighed and stored at -20˚C until required. After thawing, they were transferred to a glass homogenizer (Jencons Cat no. 361–044) in 300 μl of 0.1 M Tris/HCl buffer, pH 8, and homogenized with 20 up and down strokes. The homogenate was centrifuged at 13,000 rpm for 2 minutes to yield a supernatant for assay of aminopeptidase activity. The assays were performed at room temperature in a Fluostar Omega (BMG Labtech Gmbh, Ortenberg, Germany) microplate reader ($\lambda_{ex}$, 380 nm and $\lambda_{em}$ 460 nm) using Microfluor Black 96-well plates (ThermoFisher Scientific, Hemel Hempstead, U.K.). Each well contained 150 μl of 0.1 M Tris-HCl, pH 8, 50μl of worm supernatant and 1μl of 5 mM Lys(εDNP)-Pro-Pro-Amp and the increase in fluorescence resulting from the release of the fluorescent Pro-Pro-Amp was monitored continuously over time.

### Scoring of DAPI-stained chromatin bodies in diakinesis oocytes

Worms were dissected, fixed, and stained with DAPI as described in the immunostaining protocol. The number of DAPI-stained bodies in the two most proximal oocytes was scored from three-dimensional intact germ lines. Six DAPI-stained bodies, as observed in wild-type controls, corresponds to full chiasma formation, while 12 DAPI-stained bodies indicate complete absence of chiasmata.

### Quantification of RAD-51, OLLAS::COSA-1 and GFP::MSH-5 foci

Analysis of RAD-51 foci following immunostaining was performed as in [62]. RAD-51 agglomerates that were not detectable as single foci were categorized as stretches. At least three germ lines per genotype were used for quantification of RAD-51 foci. To quantify OLLAS::

COSA-1 and GFP::MSH-5 foci, worms were dissected in 1x M9 buffer, a coverslip was added and immediately freeze-cracked in liquid nitrogen. Slides were left in methanol at -20˚C for at least 10 minutes and 100 μl of a of 2% PFA in 0.1M $K_2HPO_4$ (pH 7.6) solution was added on each slide. Samples were covered with a parafilm square and fixation was carried out for 10 minutes at room temperature in a humid chamber. Slides were then washed three times in 1x PBST for 5 minutes each and the remaining steps of the immunostaining were performed as described above. The anti-OLLAS-tag antibody (Genscript) was diluted in 1xPBST at 1:1500. Incubation was carried out at 4˚C over-night. Detection of GFP::MSH-5 was performed by exploiting the GFP natural fluorescence, without using an anti-GFP antibody. Gonads were divided into 5 equal regions spanning the transition zone onset to late pachytene, and GFP:: MSH-5/OLLAS::COSA-1 foci were counted for each nucleus for each region.

## siRNA transfection in human cells

U2OS cells were transfected with control and *XPNPEP1* siRNA (Dharmacon/GE Healthcare) using RNAiMAX transfection reagent according to manufacturer's instruction. Cells were harvested 72 hours after siRNA transfection. The sequences of the two siRNAs to target *XPNPEP1* were: GCGACUGGCUCAACAAUUA (#1) and GGGAUUCAGGCCUAGAUUA (#2).

## Antibodies for human cell culture experiments

Antibodies used for western blot were Aminopeptidase P (Novus, NBP1-80614), Topoisomerase II Alpha (TopoGEN, TG2011-1), Tubulin (Sigma T6199) and Histone H3 (Abcam 10799). γ-H2AX (Millipore 05–636) was used for immunostaining.

## γ-H2AX immunofluorescence staining

γ-H2AX immunofluorescence staining was performed as described previously [63,64]. U2OS cells were cultured on coverslips and permeabilized with IF CSK buffer (10mM PIPES pH 6.8, 100mM NaCl, 300mM Sucrose, 3mM $MgCl_2$, 1mM EGTA, 0.5% Triton X-100, 1x protease inhibitor cocktail, 1x phosphatase inhibitor). Cells were fixed with 2% PFA at room temperature for 15 min. The coverslips were blocked in PBS with 3% BSA and Triton X-100 at room temperature for 30 minutes and incubated with primary and secondary antibodies. After washing with PBS containing 0.1% Triton X-100, the coverslips were mounted with Vectashield media.

## Protein fractionation in human cells

Chromatin fractionation was carried out as previously described [65] with a number of modifications. Cells were harvested and resuspended in CSK buffer (10mM HEPES pH 7.9, 150mM NaCl, 300mM Sucrose, 1mM $MgCl_2$, 1mM EDTA, 0.2% Triton X-100, 1x protease inhibitor cocktail, 1x phosphatase inhibitor). After incubation for 10 minutes on ice, chromatin-containing fractions were pelleted and washed in CSK buffer. Supernatant was designated the soluble fraction. The pelleted chromatin fractions were resuspended in nuclease digestion buffer and sonicated using a Bioruptor. After adding benzonase, samples were incubated at 37˚C for 30 minutes. This fraction was designated the chromatin fraction.

## Pulse field gel electrophoresis

Pulse field gel electrophoresis was performed as previously described [65]. Briefly, same number of control siRNA cells and XPNPEP1 siRNA cells were embedded in agarose plugs individually. Plugs were treated with proteinase K overnight at 50˚C. Plugs were loaded into wells of

1% agarose pulse field gel and run for 24 hours on PFGE apparatus. The gel was stained with ethidium bromide and analysed with Geldoc system.

## Supporting information

**S1 Fig. APP-1 is dispensable for chiasma formation but is required to prevent accumulation of recombination intermediates. (A)** (Top) Quantification of pairing levels using a probe for the pairing centre region of chromosome III (T17A3 cosmid) and a probe for 5S rDNA locus on chromosome V. Gonads were divided into six equal regions and pairing was quantified in each region. Pairing occurs normally in *app-1* mutants. (Bottom) Representative images of pachytene nuclei labelled with FISH probes. The number of nuclei counted for each region were (WT, *app-1*): Zone 1 (77, 104), zone 2 (56, 137), zone 3 (98, 196), zone 4 (209, 173), zone 5 (198, 162), and zone 6 (89, 87). **(B)** Representative images of pachytene nuclei in WT and *app-1* worms stained with α-SYP-1 (central component of SC) and α-HTP-1/2 (axial element components) showing normal synapsis. **(C)** RAD-51 foci accumulate in germ lines of *app-1(fq96)* and *app-1(fq96); spo-11* mutants. Graphs display the regions along the germ line on the X axis (see Fig 1D for image of germ line with zones) and the percentage of nuclei with a given number of RAD-51 foci as indicated in the color key on the Y axis. The number of nuclei analysed per genotype and zone were: WT (372, 214, 144, 173, 129, 120, 111), *app-1 (fq96)* (257, 126, 110, 97, 103, 80, 77), *spo-11* (215, 180, 164, 147, 138, 102, 104), *app-1(fq96); spo-11* (247, 161, 129, 132, 114, 109, 74). Examples show projections of whole germ lines of the indicated genotypes stained with anti-RAD-51 antibodies and DAPI. **(D)** Quantification of embryonic lethality and RAD-51 foci in worms expressing a wild-type *app-1* transgene [*fqSi19*] in the *app-1(tm1715)* mutant background. Note that *app-1(tm1715); fqSi19[app-1 WT]* worms display low lethality and normal levels of RAD-51 foci. Number of embryos scored: WT 3148 embryos from 13 worms, *app-1 (tm1715)* 2073 embryos from 17 worms, and *app-1(tm1715) fqSi19[app-1 WT]* 4006 embryos from 19 worms and statistical significance was calculated by a one-way ANOVA test (**** P<0.0001). The number of nuclei analysed per zone for RAD-51 foci analysis were: 186, 166, 160, 160, 165, 111, 139. **(E)** Quantification of RAD-51 foci in mutants of indicated genotypes, note accumulation of RAD-51 foci in *app-1 (ttTi14848)* and *app-1(ttTi14848); spo-11*. The number of nuclei analysed per genotype and zone were: *app-1(ttTi14848)* (179, 172, 120, 171, 206, 98, 91) and *app-1(ttTi14848); spo-11* (151, 192, 207, 232, 213, 186, 135). See S1 Table for underlaying numerical data of graphs. (TIF)

**S2 Fig. Proliferating germ cells in *app-1* mutants show increased levels of RPA-1 and TOP-2. (A)** Undifferentiated germ cells in *app-1* mutant germlines display increased numbers of RPA-1 foci. Projections of deconvolved images from the mitotic compartment of the germ line from WT and *app-1(tm1715)* mutant worms carrying an *rpa-1:YFP* transgene. Note increased numbers of RPA::YFP foci (arrow heads) in *app-1(tm1715)* mutants. Graph shows % of proliferating germ-cell nuclei positive for RPA-1 per germline (10 WT and 13 *app-1(tm1715)* germ lines were analysed. **(B)** Undifferentiated germ cells in *app-1* mutant germ lines display increased levels of topoisomerase II. Non-deconvolved projections of mitotic germline nuclei of worms carrying a 3XFLAG tag on the endogenous *top-2* locus. Images were acquired with the same exposure and adjusted with the same settings to allow direct comparison of the intensity of TOP-2 staining (anti-FLAG antibodies) in wild-type and *app-1(tm1715)* mutant germ lines. Note increased levels of TOP-2 in *app-1(tm1715)* mutant germ line. Right-hand panels show deconvolved images of the same germ lines. Scale bar = 5 μm. See S1 Table for underlaying numerical data of graphs. (TIF)

**S3 Fig. *app-1* mutants do not display increased sensitivity to IR exposure.** IR dose shown on X axis and viability, assessed 24 hours post irradiation, on the Y axis. *app-1* mutants respond similarly to WT worms, in contrast with the hypersensitivity of *brc-1* mutants used as control. Bars represent standard error of the mean (SEM). Number of embryos scored (WT, *brc-1*, *app-1*): 0 Gy (778, 753, 272), 40 Gy (643, 629, 565), 80 Gy (553, 452, 442), and 120 Gy (541, 703, 334). See S1 Table for underlaying numerical data of graphs.
(TIF)

**S4 Fig. PARPs contribute to RAD-51 foci accumulation in *app-1* mutants. (A)** Quantification of RAD-51 in germ lines of *parp-1* and *parp-2* single mutants show normal levels of RAD-51 foci. The number of nuclei used for RAD-51 foci quantification per genotype and zone were: *parp-1* (211, 181, 182, 199, 143, 115, 160) and *parp-2* (149, 210, 167, 160, 141, 174, 164). (B-C) Examples of RAD-51 staining in pachytene nuclei of indicated genotypes from graphs shown in panel A and in Fig 4B. See S1 Table for underlaying numerical data of graphs.
(TIF)

**S5 Fig. Non-homologous chromosome fusions are observed in *app-1; spo-11* oocytes.** Projections of diakinesis nuclei labelled with FISH probes to visualize chromosomes III (red) and V (green). In WT and *app-1* oocytes each probe is associated with a single bivalent, demonstrating attachment of homologous chromosomes by chiasmata. *spo-11* oocytes lack chiasmata and therefore FISH probes label two separated chromatin bodies. Bottom panel displays three examples of oocytes from *app-1; spo-11* double mutants labelled with the same FISH probes. Oocyte on the left-hand side panel displays 6 DAPI-stained bodies and each FISH probe is associated with a single chromatin mass, demonstrating attachments between homologous chromosomes. In the middle panel only four chromatin bodies are present and the probe for chromosome III is found on two different bodies, suggesting the presence of attachments between non-homologous chromosomes. The oocyte on the right-hand panel displays 5 chromatin masses; the probe for chromosome III is found in two of them, which according to size and shape may represent an isolated univalent plus a univalent fused to another unlabelled chromosome; the probe for chromosome V is found in a larger chromatin mass that also suggests the presence of attachments between non-homologous chromosomes. Scale bar = 5 μm.
(TIF)

**S6 Fig. Removal of DVC-1 does not restore chiasma formation in *spo-11* mutants.** Quantification of the number of DAPI-stained bodies in diakinesis oocytes of indicated genotypes and examples shown on right-hand panel. Note that both *spo-11* and *spo-11 dvc-1* mutants display mostly 12 DAPI-stained bodies, indicating a failure in chiasma formation. Number of diakinesis nuclei scored: WT (20), *dvc-1* (22), *spo-11 dvc-1* (62), and *spo-11* (45). Scale bar = 5 μm.
(TIF)

**S1 Table. Underlaying numerical data of graphs.**
(XLSX)

## Acknowledgments

We thank Karl Payne and Tony Turner (University of Leeds) for generating reagents to measure the catalytic activity of APP-1 and Aimee Jaramillo-Lambert for sharing *C. elegans* strains. REI acknowledges the valuable advice provided by Ian Hope (University of Leeds).

## Author Contributions

**Conceptualization:** Nicola Silva, R. Elwyn Isaac, Simon J. Boulton, Enrique Martinez-Perez.

**Data curation:** Nicola Silva, Maikel Castellano-Pozo, Kenichiro Matsuzaki, Consuelo Barroso, R. Elwyn Isaac, Simon J. Boulton, Enrique Martinez-Perez.

**Formal analysis:** Nicola Silva, Maikel Castellano-Pozo, Kenichiro Matsuzaki, Consuelo Barroso, R. Elwyn Isaac, Simon J. Boulton, Enrique Martinez-Perez.

**Funding acquisition:** Nicola Silva, Darren R. Brooks, R. Elwyn Isaac, Simon J. Boulton, Enrique Martinez-Perez.

**Investigation:** Nicola Silva, Maikel Castellano-Pozo, Kenichiro Matsuzaki, Consuelo Barroso, Monica Roman-Trufero, Hannah Craig, Darren R. Brooks, R. Elwyn Isaac, Simon J. Boulton, Enrique Martinez-Perez.

**Supervision:** R. Elwyn Isaac, Simon J. Boulton, Enrique Martinez-Perez.

**Writing – original draft:** Nicola Silva, R. Elwyn Isaac, Simon J. Boulton, Enrique Martinez-Perez.

**Writing – review & editing:** Nicola Silva, Maikel Castellano-Pozo, Kenichiro Matsuzaki, Consuelo Barroso, Monica Roman-Trufero, Hannah Craig, Darren R. Brooks, R. Elwyn Isaac, Simon J. Boulton, Enrique Martinez-Perez.

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
