## [Decision Letter · Decision Letter 0]

18 Oct 2021

Dear Dr Martinez-Perez,

Thank you very much for submitting your Research Article entitled 'Proline-specific aminopeptidase P prevents replication-associated genome instability' to PLOS Genetics.

The manuscript was fully evaluated at the editorial level and by independent peer reviewers. The reviewers appreciated the attention to an important topic but identified some minor concerns that we ask you address in a revised manuscript.

We therefore ask you to modify the manuscript according to the review recommendations. Your revisions should address the specific points made by each reviewer. In particular, please make the minor corrections and clarifications to the text requested by the reviewers, include the statistics missing in some of the figures, address the issue with the large RAD-51 aggregates raised by Reviewers 1 and 3, and consider including RPA-1 analysis (while the latter is not required, if readily available/possible it would further strengthen the manuscript).

[LINK]

Yours sincerely,

Mónica P. Colaiácovo

Associate Editor

PLOS Genetics

Gregory P. Copenhaver

Editor-in-Chief

PLOS Genetics

Reviewer's Responses to Questions

**Comments to the Authors:**

Reviewer #1: The manuscript by Silva et al. describes the unexpected finding that Proline-specific aminopeptidase APP-1, plays a role in preventing replication stress. Using genetics, cell biology and biochemical approaches the authors show convincingly that APP-1 prevents replication stress in C. elegans and extend their findings in human tissue culture. I have only minor suggestions to improve the manuscript:

1. The authors look at RAD-51 in the C. elegans germline and note that there are large aggregates of RAD-51 – mutant and wild type dont look any different to me based on the limited images shown in the supplement. Could the authors quantify this or direct the reader to what they consider aggregates?

2. In Figure 3C: please add “IR” to the figure, not just the minutes.

3. Analysis of app-1; spo-11 double mutants suggests that the lesions produced by the app-1 mutant can serve as substrates for HR and crossover formation. Is embryonic viability improved in the app-1; spo-11 double mutant?

4. Please add statistics to the figures. Specifically, in figure 4 analyses of the genetic interaction between app-1 and parp-1/2 should be analyzed by ONE WAY ANOVA, but statistics are missing from some other figures as well.

5. In the introduction on page 4, the authors state that there are “two pathways” for repair of DSBs – I recommend qualifying this to say two major/main pathways, as other pathways have been shown to play a role in the germ line.

Reviewer #2: The paper by Silva at al. is well written and the conclusions are generally supported by the experimental evidence. In short, the authors find that the APP1 aminopeptidase is require for genome maintenance in the C. elegans germ line. Evidence is provided that this due to defective replication. Such defect results in increased RAD-51 focus formation, which the authors characterize reasonably well (see below). In the worm, it is known that excessive DNA double stand breaks, can bypass the need for the SPO-11 nuclease (required for meiotic double strand breaks), and this is what the authors also observe due to excessive double strand breakage in app1 mutants. They study this bypass is detail. The authors also provide evidence of increased genome instability upon APP1 depletion in mammalian cells.

In summary, in all the authors providing a solid first pass analysis of APP1 phenotypes.

We do not learn what the targets of APP1 are. This is fine, as the authors clearly acknowledging this in their discussion. All in all, it appears likely that we are looking at a DNA replication defect associated with APP1 deficiency. Experimentation, is not at the state of the art, neither in the worm nor in mammalian cells to further dissect this. As to this, in the C. elegans germ line, I think there would be a case for measuring the accumulation of RPA-1 foci, which by marking single stranded DNA are more direct indicator of DNA replication defects. Recent papers describe RPA::GFP fusions etc etc … Generally, neither in the worm, nor in U2OS cells attempts are made to study the replication defect in any detail. In the worm for instance, DNA replication defects, when associated with checkpoint activation led to prolonged embryonic cell cycles especially of the P1 cells. Specialist papers, on further dissecting replication defects have been published. Of course, there is 1001 things that could be done in mammalian cells.

All in all, the data is solid, this is a genetic analysis for a genetics journal. Outing myself as a geneticist, I overall support publication.

Reviewer #3: The manuscript of Silva et al provides detailed analysis of the role for aminopeptidase APP-1 in the C. elegans germ line showing that APP-1 prevents the accumulation of replication stress. They also show that this function of APP-1 is conserved in human cells and prevents the accumulation of nuclear-associated topoisomerase II. The data convincing show the presence of mitotic DNA damage and SPO-11 independent DNA damage. Strikingly this is the first mutant which shows such a dramatic increase in damage that can be repaired with high fidelity through the HR crossover pathway, leading to suppression of the crossover due to lack of spo-11. Overall, the data are solid and the writing is clear and the paper will be of broad interest to the DNA repair, cancer, and meiosis communities. There are minor issues with the figures and text that need to be addressed prior to publication and a few experiments that would greatly enhance this first description of the app-1 mutants. The only major issue is to clarify whether the cell cycle is simply slower or the cells arrest in response to damage since the authors propose in the discussion that APP-1 prevents the formation of replication induced damage rather than functioning in the repair.

• A description of the developmental defects quantified in Figure 1 should be provided, especially since the next paragraph states they are reminiscent of DNA damage deficient worms.

•. Are the brood, embryonic lethality, and developmental defects from app-1 homozygous stocks or from heterozygous mothers or which generation off a balancer? This information is not in the Methods and should be provided in the Figure legend. This is critical if app-1 is a mutator, in which case, providing info on zygotic vs maternal effect lethality should be included.

•. Is there an increase in apoptosis in these mutants. Since the suggestion is that all the damage is repaired, one might expect there is not, this could distinguish it from other mutants and also address whether checkpoints are activated.

• Are the enlarged proliferative nuclei arresting in the cell cycle due to DNA damage that can be reported by activation of HUS-1-dependent checkpoint? The evidence presented is convincing that the cells are arrested and not further responded to HU because they are not cycling as quickly (though this is not explicitly stated), but the cells could be arrested in S, G2, or at the onset of M. Is there any to indicate which phase they are in? This would be a nice addition to the text.

• Is there also no effect on survival after HU until the higher doses?

•. The accumulation of TopoII in the U2OS cells could be a consequence of a cell cycle defect: increasing the number of cells in S, G2, and M relative to G1. The cells should be sorted by DNA contents and then analyzed for TopoII in G2 cells.

• Figure S2. I am not convinced by that staining shown that Topoisomerase II levels are different in the mutants because the wild type image seems “fuzzy”. Also, wouldn’t you expect the accumulation in the enlarged cells but maybe not the normal sized cells?

• The stretches of RAD-51 that are described for app-1 mutants should be pointed to in the figure (S1C) since it appears to my eyes that the WT control also has a couple stretches, but maybe I am misinterpreting what they mean. Figure S1C should also say which “zone” the cells are from.

• There are a few unintuitive intellectual leaps that would help the reader better understand the phenotypes, most crticially explaining why app-1 mutants would be seemingly resistant to HU which is thought to induce replicative stress. Is it possible that APP-1 is acting through the same pathway?

• Figure 1C: In the image of tm1715, it appears that one of the bivalents is separating. Is this typical of the mutant?

•.Why are the n values given in Suppl tables except for Fig 3C, 4A? It seems these could be included in the figure legends throughout. Either way, consistency would be preferred.

• In Figure S5, bottom right, the chromosome III probes appear to be on univalents based on size of the left-most DAPI spot; the right-most spot appears smaller as well and in close juxtaposition to a bivalent-like structure rather than part of it. Also, one cannot say that NHEJ is producing the observed fusions (as the title of the figure and text p 17 suggests); it could be MMEJ (or other pathways) as well.

•. Does app-1 act as a mutator? Just because there is a excessive DNA damage, if it is all repaired, it may not cause genome instability per se but rather just cell cycle delay.

• Discussion: “point to APP-1 acting to prevent the onset of replication-related DNA damage rather than being required for its repair”. It might be further convincing to add that the synthetic lethality with parp1/2 support this model.

•.Discussion of the 3 models for APP-1 (at least models 1 and 2) appear vague without any analysis of whether known proteins involved in these processes might fit into these categories.

Textual:

Summary: Add comma after “In this manuscript…”

Results line 1: Add comma after “C. elegans”

Fig 1E legend. Enzymatic “assay” not “essay”

Figure 3C, it would be helpful to indicate in the figure that this is IR (perhaps in Y axis add “post-IR)

Results: “The resulting app-1H392A H496A transgene (called app-1CD hereafter) was

crossed to worms homozygous for the app-1(ttTi4848) allele, which carries a

transposon insertion on the third exon of app-1 (Figure 1A) and that similar to worms

homozygous for the app-1(tm1715) and app-1(fq96) alleles also accumulate SPO-11-

independent RAD-51 foci (Figure S1F).” This is confusing. Can you say: “The resulting app-1H392A H496A transgene (called app-1CD hereafter) was tested for its ability to rescue the app-1 mutant app-1(ttTi4848) that also accumulates SPO-11-independent RAD-51 foci (Figure S1F).”

Page 10, Lines 1,2: “wild-type” (with hyphen) when used as an adjective

Page 10, line 7. “germline” (one word) as adjective; “germ line” when noun

“program” not “programme”

p. 15 middle of the page: at times it is “mre-11; app-1” and others ”app-1; mre-11” (also com-1).

“app-1; spo-11double mutants accumulated extensive MSH-5 and COSA-1 foci from early pachytene (Figure 6B).” This is confusing because it suggests that early pachytene is somehow different from wild type, which is not the case based on the quanitifications in the figure.

Figure 7C is discussed prior to the rest of figure 7.

British vs American English of analyse, ionizing. Check with journal which is preferred.

One would expect that the mitotic damage is SPO-11-independent: “This possibility is further suggested by accumulation of SPO-11-independent RAD-51 foci in mitotic and meiotic nuclei”. Maybe should read “… accumulation of mitotic and SPO-11-independent meiotic RAD-51 foci..”

P. 19 2nd line from bottom, last word should be “repair” not “repairing”

**Have all data underlying the figures and results presented in the manuscript been provided?**

Reviewer #1: Yes

Reviewer #2: Yes

Reviewer #3: Yes

PLOS authors have the option to publish the peer review history of their article (what does this mean?). If published, this will include your full peer review and any attached files.

Reviewer #1: No

Reviewer #2: No

Reviewer #3: No

---

## [Editor Report · Decision Letter 1]

10 Jan 2022

Dear Dr Martinez-Perez,

We are pleased to inform you that your manuscript entitled "Proline-specific aminopeptidase P prevents replication-associated genome instability" has been editorially accepted for publication in PLOS Genetics. Congratulations!

Yours sincerely,

Mónica P. Colaiácovo

Associate Editor

PLOS Genetics

Gregory Copenhaver

Editor-in-Chief

PLOS Genetics

Comments from the reviewers (if applicable):

**Data Deposition**

http://datadryad.org/submit?journalID=pgenetics&manu=PGENETICS-D-21-01257R1

**Press Queries**

---

## [Editor Report · Acceptance letter]

21 Jan 2022

PGENETICS-D-21-01257R1 

Proline-specific aminopeptidase P prevents replication-associated genome instability 

Dear Dr Martinez-Perez, 

We are pleased to inform you that your manuscript entitled "Proline-specific aminopeptidase P prevents replication-associated genome instability" has been formally accepted for publication in PLOS Genetics! Your manuscript is now with our production department and you will be notified of the publication date in due course.

With kind regards,

Livia Horvath

PLOS Genetics

On behalf of:
